# Barren plateaus in quantum neural network training landscapes

Jarrod R. McClean[1], Sergio Boixo [1], Vadim N. Smelyanskiy[1], Ryan Babbush[1] & Hartmut Neven[1]

Many experimental proposals for noisy intermediate scale quantum devices involve training a parameterized quantum circuit with a classical optimization loop. Such hybrid quantum-classical algorithms are popular for applications in quantum simulation, optimization, and machine learning. Due to its simplicity and hardware efficiency, random circuits are often proposed as initial guesses for exploring the space of quantum states. We show that the exponential dimension of Hilbert space and the gradient estimation complexity make this choice unsuitable for hybrid quantum-classical algorithms run on more than a few qubits. Specifically, we show that for a wide class of reasonable parameterized quantum circuits, the probability that the gradient along any reasonable direction is non-zero to some fixed precision is exponentially small as a function of the number of qubits. We argue that this is related to the 2-design characteristic of random circuits, and that solutions to this problem must be studied.

---

[1] Google Inc., 340 Main Street, Venice, CA 90291, USA. Correspondence and requests for materials should be addressed to J.R.M. (email: jmcclean@google.com) or to S.B. (email: boixo@google.com) or to V.N.S. (email: smelyan@google.com)

Rapid developments in quantum hardware have motivated advances in algorithms to run in the so-called noisy intermediate scale quantum (NISQ) regime[1]. Many of the most promising application-oriented approaches are hybrid quantum–classical algorithms that rely on optimization of a parameterized quantum circuit[2–8]. The resilience of these approaches to certain types of errors and high flexibility with respect to coherence time and gate requirements make them especially attractive for NISQ implementations[3,9–11].

The first implementation of such algorithms was developed in the context of quantum simulation with the variational quantum eigensolver[2,3]. This algorithm has been successfully demonstrated on a number of experimental setups with extensions to excited states and other forms of incoherent error mitigation[2,9,12–16]. Since then, the quantum approximate optimization algorithm was developed in a similar context to address hard optimization problems[5,17–19]. This algorithm has also been demonstrated on quantum devices[20]. These approaches have even been extended to both quantum machine learning and error correction[6,7,20–23].

While the precise formulation of these methods and their domains of applicability differ considerably, they typically tend to rely upon the optimization of some parameterized unitary circuit with respect to an objective function that is typically a simple sum of Pauli operators or fidelity with respect to some state. This framework is reminiscent of the methodology of classical neural networks[23,24]. As with any non-linear optimization, the choice of both the parameterization and the initial state is important. In quantum simulation, there is often a choice inspired by physical domain knowledge[3,17,25–29]. However, in all domains of applicability, there have been implementations that utilize parametrized random circuits of varying depth[7,13,21,23,30]. Within quantum simulation that approach has been referred to as a "hardware efficient ansatz"[13]. This is in contrast to the previous proposals, such as the variational quantum eigensolver[2,3,9], which used parametrized structured circuits inspired by the problem at hand, such as unitary coupled cluster.

When little structure is known about the problem, or constraints of the existing quantum hardware may prevent utilizing that structure, choosing a random implementable circuit seems to provide an unbiased choice. One might also expect, based on recent experimental designs for "quantum supremacy", that random quantum circuits are a powerful tool for such a task[31]. Also, despite concerns about gradient-based methods in classical deep neural networks[32–34], they are successful[24], even if using random

initialization[33,35]. However, in the quantum case one must remember that the estimation of even a single gradient component will scale as $O(1/\varepsilon^\alpha)$ for some small power $\alpha$[36] as opposed to classical implementations where the same is achieved in $O(\log(1/\varepsilon))$ time, where $\varepsilon$ is the desired accuracy in the gradient that is inevitably tied to its magnitude.

We will present results related to random quantum circuits in the context of the exponential dimension of Hilbert space and gradient-based hybrid quantum–classical algorithms. A cartoon depiction of this is given in Fig. 1. We show that for a large class of random circuits, the average value of the gradient of the objective function is zero, and the probability that any given instance of such a random circuit deviates from this average value by a small constant $\varepsilon$ is exponentially small in the number of qubits. This can be understood in the geometric context of concentration of measure[37–39] for high-dimensional spaces. When the measure of the space concentrates in this way, the value of any reasonably smooth function will tend towards its average with exponential probability, a fact made formal by Levy's lemma[40]. In our context, this means that the gradient is zero over vast reaches of quantum space. The region where the gradient is zero does not correspond to local minima of interest, but rather an exponentially large plateau of states that have exponentially small deviations in the objective value from the average of the totally mixed state. We argue that the depth of circuits which achieve these undesirable properties are modest, requiring only $O(n^{1/d})$ depth circuits on a $d$ dimensional array, and numerically evaluate the constant factors one expects to encounter for small instances of this kind. While our results highlight the importance of avoiding random initialization in parametric circuit approaches, they do not discount the value of random quantum circuits in other applications such as information security or demonstrations of quantum supremacy. We close with an outlook on how this result should shape strategies in ansatz design for scaling to larger experiments.

## Results

**Gradient concentration in random circuits.** We will discuss random parameterized quantum circuits (RPQCs)

$$U(\boldsymbol{\theta}) = U(\theta_1, ..., \theta_L) = \prod_{l=1}^{L} U_l(\theta_l) W_l, \tag{1}$$

where $U_l(\theta_l) = \exp(-i\theta_l V_l)$, $V_l$ is a Hermitian operator, and $W_l$ is a generic unitary operator that does not depend on any angle $\theta_l$. Circuits of this form are a natural choice due to a straightforward evaluation of the gradient with respect to most objective functions and have been introduced in a number of contexts already[26,41]. Consider an objective function $E(\theta)$ expressed as the expectation value over some Hermitian operator $H$,

$$E(\boldsymbol{\theta}) = \langle 0|U(\boldsymbol{\theta})^\dagger H U(\boldsymbol{\theta})|0\rangle. \tag{2}$$

When the RPQCs are parameterized in this way, the gradient of the objective function takes a simple form:

$$\partial_k E \equiv \frac{\partial E(\boldsymbol{\theta})}{\partial \theta_k} = i\langle 0|U_-^\dagger \left[V_k, U_+^\dagger H U_+\right] U_-|0\rangle, \tag{3}$$

where we introduce the notations $U_- \equiv \prod_{l=0}^{k-1} U_l(\theta_l) W_l$, $U_+ \equiv \prod_{l=k}^{L} U_l(\theta_l) W_l$, and henceforth drop the subscript $k$ from $V_k \to V$ for ease of exposition. Finally, we will define our RPQCs $U(\boldsymbol{\theta})$ to have the property that for any gradient direction $\partial_k E$ defined above, the circuit implementing $U(\boldsymbol{\theta})$ is sufficiently random such that either $U_-$, $U_+$, or both match the Haar

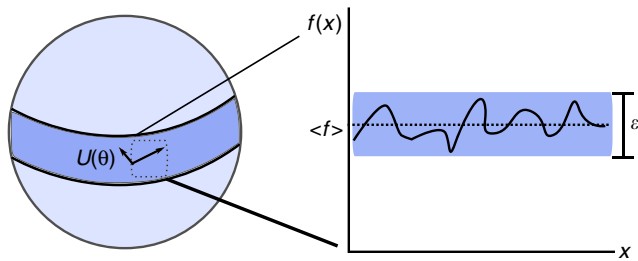

**Fig. 1** Cartoon of concentration of quantum observables. The sphere depicts the phenomenon of concentration of measure in quantum space: the fraction of states that fall outside a fixed angular distance from zero along any coordinate decreases exponentially in the number of qubits[40]. This implies a flat plateau where observables concentrate on their average over Hilbert space and the gradient is exponentially small. The fact that only an exponentially small fraction of states fall outside of this band means that searches resembling random walks will have an exponentially small probability of exiting this "barren plateau"

distribution up to the second moment, and the circuits $U_-$ and $U_+$ are independent.

Our results make use of properties of the Haar measure on the unitary group $d\mu_{\text{Haar}}(U) \equiv d\mu(U)$, which is the unique left- and right-invariant measure such that

$$\int_{U(N)} d\mu(U)f(U) = \int d\mu(U)f(VU) = \int d\mu(U)f(UV), \quad (4)$$

for any $f(U)$ and $V \in U(N)$, where the integration domain will be implied to be $U(N)$ when not explicitly listed. While this property is valuable for proofs, quantum circuits that exactly achieve this invariance generically require exponential resources. This motivates the concept of unitary $t$-designs[42–44], which satisfy the above properties for restricted classes of $f(U)$, often requiring only modest polynomial resources. Suppose $\{p_i, V_i\}$ is an ensemble of unitary operators, with unitary $V_i$ being sampled with probability $p_i$. The ensemble $\{p_i, V_i\}$ is a $t$-design if

$$\sum_i p_i V_i^{\otimes t} \rho (V_i^\dagger)^{\otimes t} = \int d\mu(U) U^{\otimes t} \rho (U^\dagger)^{\otimes t}. \quad (5)$$

This definition is equivalent to the property that if $f(U)$ is a polynomial of at most degree $t$ in the matrix elements of $U$ and at most degree $t$ in the matrix elements of $U^*$, then averaging over the $t$-design $\{p_i, V_i\}$ will yield the same result as averaging over the unitary group with the respect to the Haar measure.

The average value of the gradient is a concept that requires additional specification because, for a given point, the gradient can only be defined in terms of the circuit that led to that point. We will use a practical definition that leads to the value we are interested in, namely

$$\langle \partial_k E \rangle = \int dU p(U) \partial_k \langle 0 | U(\boldsymbol{\theta})^\dagger H U(\boldsymbol{\theta}) | 0 \rangle, \quad (6)$$

where $p(U)$ is the probability distribution function of $U$. A review on the properties of products of independent random matrices can be found in ref.[45]. The assumptions of independence and at least one of $U_-$ or $U_+$ forming a 1-design in our RPQCs implies that $\langle \partial_k E \rangle = 0$, as shown in the Methods.

Levy's lemma informs our intuition about the the expected variance of this quantity through simple geometric arguments. In particular, Haar random unitaries on $n$ qubits will output states

uniformly in the $D = 2^n - 1$ dimensional hypersphere. The derivative with respect to the parameters $\theta$ is Lipschitz continuous with some parameter $\eta$ that depends on the operator $H$. Levy's lemma then implies that the variance of measurements will decrease exponentially in the number of qubits. This intuition may be made more precise through explicit calculation of the variance, which is done in more detail in the Methods. The result to first order is

$$\text{Var}[\partial_k E] \approx \begin{cases} -\frac{\text{Tr}(\rho^2)}{(2^{2n}-1)} \text{Tr} \left\langle \left[ V, u^\dagger H u \right]^2 \right\rangle_{U_+} \\ -\frac{\text{Tr}(H^2)}{(2^{2n}-1)} \text{Tr} \left\langle \left[ V, u\rho u^\dagger \right]^2 \right\rangle_{U_-} \\ \frac{1}{2^{(3n-1)}} \text{Tr}(H^2)\text{Tr}(\rho^2)\text{Tr}(V^2) \end{cases} \quad (7)$$

where the notation $\langle f(u) \rangle_{U_x}$ indicates the average with $u$ drawn from $p(U_x)$, and the first case corresponds to $U_-$ being a 2-design and not $U_+$, the second to $U_+$ being a 2-design but not $U_-$, and the third to both $U_+$ and $U_-$ being 2-designs. We emphasize the fact that this variance depends at most on polynomials of degree 2 in $U$ and polynomials of degree 2 in $U^*$. Whereas a unitary 2-design will exhibit the correct variance[43,46], a unitary 1-design will exhibit the correct average value, but not necessarily the variance. As a result, if a circuit is of sufficient depth that for any $\partial_k E$, either $U_-$ or $U_+$ forms a 2-design, then with high probability one will produce an ansatz state on a barren plateau of the quantum landscape, with no interesting search directions in sight.

From these results, it is clear that only either $U_+$ or $U_-$ needs to be sufficiently random to poison the gradient for the remainder of the circuit. For example, while it is somewhat unintuitive, even the first element of a circuit, $k = 1$, will have a vanishing gradient due to the circuit following it, $U_+$. Additionally, we see that there is no detailed dependence on the structure of $V_k$, other than the rate at which they help randomize the circuit, determining at what depth one expects to find an approximate 2-design.

**Numerical simulations**. The previous section shows that for reasonable classes of RPQCs at a sufficient number of qubits and depth, one will end up on a barren plateau. Here we verify this result for even modest depth one-dimensional (1D) random circuits with numerical simulations. This helps to clarify the rate of concentration for realistic circuits and shows the transition as

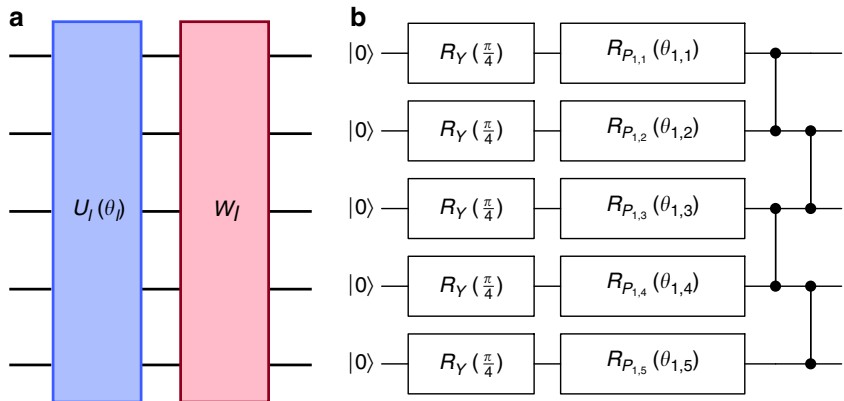

**Fig. 2** Structure of quantum circuits. **a** The generic subunit of circuits we study in this work, with a parameterized component $U_l(\theta_l)$ and non-parameterized unit $W_l$ for each layer $l$. **b** Example schematic of the 1D random circuits used in our numerical experiments. The circuit begins with $R_Y\left(\frac{\pi}{4}\right)$ gates applied to all qubits followed by a specified number of layers of randomly chosen Pauli rotations applied to each qubit and then a 1D ladder of controlled $Z$ gates. The initial $R_Y\left(\frac{\pi}{4}\right)$ gates are not repeated in each layer. The indices $i$ and $j$ in $\theta_{i,j}$ index the layer and qubit, respectively. For each layer and qubit $P_{i,j} \in \{X, Y, Z\}$ and $\theta_{i,j} \in [0, 2\pi)$ are sampled independently

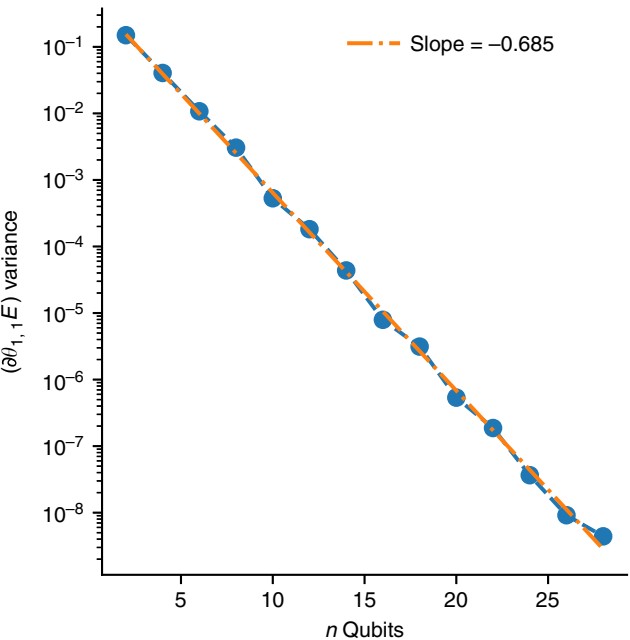

**Fig. 3** Exponential decay of variance. The sample variance of the gradient of the energy for the first circuit component of a two-local Pauli term $\left(\partial_{\theta_{1,1}} E\right)$ plotted as a function of the number of qubits on a semi-log plot. As predicted, an exponential decay is observed as a function of the number of qubits, $n$, for both the expected value and its spread. The slope of the fit line is indicative of the rate of exponential decay as determined by the operator

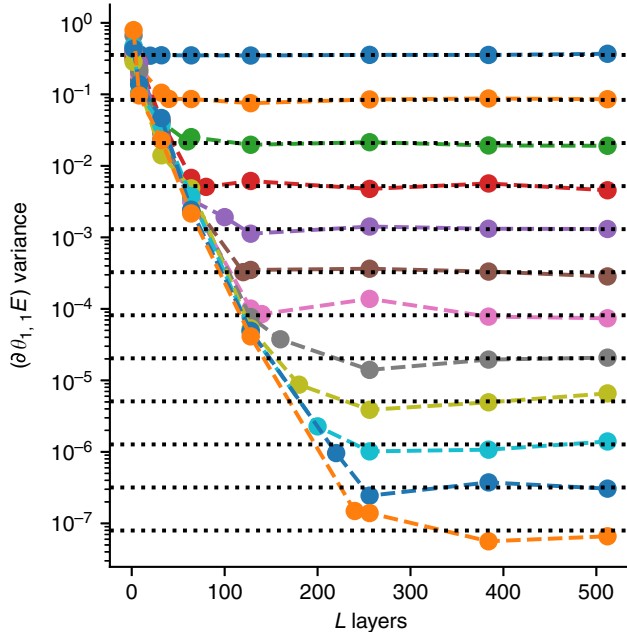

**Fig. 4** Convergence to 2-design limit. Here we show the sample variance of the gradient of the energy for the first circuit component of a two-local Pauli term $\left(\partial_{\theta_{1,1}} E\right)$ plotted as a function of the number of layers, $L$, in a 1D quantum circuit. The different lines correspond to all even numbers of qubits between 2 and 24, with 2 qubits being the top line, and the rest being ordered by qubit number. The dotted black lines depict the 2-design asymptotes for this Hamiltonian as determined by our analytic results. This shows the convergence of the second moment as a function of the number of layers to a fixed value determined by the number of qubits

the circuit grows in length from a single layer to a circuit demonstrating statistics analogous to a 2-design.

The circuits and objective functions used in our numerical experiments begin with a layer of $R_Y(\pi/4) = \exp(-i\pi/8\, Y)$ gates to prevent $X$, $Y$, or $Z$ from being an especially preferential direction with respect to gradients. Then, the circuit proceeds by a number of layers. Each layer consists of a parallel application of single qubit rotations to all qubits, given by $R_P(\theta)$ where $P \in \{X, Y, Z\}$ is chosen with uniform probability and $\theta \in [0, 2\pi)$ is also chosen uniformly. This layer is followed by a layer of 1D nearest neighbor controlled phase gates, as in Fig. 2. Thus, the number of angles is the number of qubits times the number of layers.

The objective operator $H$ is chosen to be a single Pauli ZZ operator acting on the first and second qubits, $H = Z_1 Z_2$. The gradient is evaluated with respect to the first parameter, $\theta_{1,1}$. This simple choice helps to extract the exponential scaling. As complex objectives can be written as sums of these operators, the results for large objectives can be inferred from these numbers. Moreover, it is clear that for any polynomial sum of these operators, the exponential decay of the signal in the gradient will not be circumvented.

From Fig. 3 we see that for a single 2-local Pauli term, both the expected value of the gradient and its spread decay exponentially as a function of the number of qubits even when the number of layers is a modest linear function. Empirically for our linear connectivity, we see that value is about $10n$ where $n$ is the number of qubits, following the expected scaling of $O(n^{1/d})$ where $d$ is the dimension of the connectivity. For empirical reference, the expected gate depth in a chemistry ansatz such as unitary coupled cluster is at least $O(n^3)$, meaning that if the initial parameters were randomized, this effect could be expected on less than 10 orbitals, a truly small problem in chemical terms. We also observe in Fig. 4 that as the number of layers increases, there is a transition to a 2-design where the variance converges. This leads

to a distinct plateau as the circuit length increases, where the height of the plateau is determined by the number of qubits. An additional example with an objective function defined by projection on a target state is provided as Supplementary Figures 1 and 2, showing the rapid decay of variance and similar plateaus as a function of circuit length. These results substantiate our conclusion that gradients in modest-sized random circuits tend to vanish without additional mitigating steps.

**Contrast with gradients in classical deep networks**. Finally, we contrast our results with the vanishing (and exploding) gradient problem of classical deep neural networks[32–34,47]. At least two key differences are present in the quantum case: (i) the different scaling of the vanishing gradient and (ii) the complexity of computing expected values.

The gradient in a classical deep neural network can vanish exponentially in the number of layers[32,33], while in a quantum circuit the gradient may vanish exponentially in the number of qubits, as shown above. In the classical case, the gradient for a weight in a neuron depends on the sum of all the paths connecting that neuron to the output, and when the weights are initialized with random values the paths have random signs which cancels the signal[32]. The number of paths is exponential in the number of layers. In the quantum case, the number of paths is exponential in the number of gates, and also have random signs[31]. The gradient saturates to an exponential in the number of qubits because the output state is normalized.

The estimation of the gradient for each training batch for a classical neural network is limited by machine precision and scales with $O(\log(1/\varepsilon))$. Even if the gradient is small, as long as it is consistent enough between batches, the method may eventually

succeed. On a quantum device, the cost of estimating the gradient scales as $O(1/\varepsilon^{\alpha})$[36]. For any number of measurements much lower than $1/||g||^{\alpha}$, where $||g||$ is the norm of the gradient, a gradient-based optimization will result in a random walk. By concentration of measure, a random walk will have exponentially small probability of exiting the barren plateau. As a result, gradient descent without some additional strategy cannot circumvent this challenge on a quantum device in polynomial time.

## Discussion

We have seen both analytically and numerically that for a wide class of random quantum circuits, the expected values of observables concentrate to their averages over Hilbert space and gradients concentrate to zero. This represents an interesting statement about the geometry of quantum circuits and landscapes related to hybrid quantum–classical algorithms. More practically, it means that randomly initialized circuits of sufficient depth will find relatively little utility in hybrid quantum–classical algorithms.

Historically, vanishing gradients may have played a role in the early winter of deep neural networks[32,34,47]. However, multiple techniques have been proposed to mitigate this problem[24,35,48,49], and the amount of training data and computational power available has grown substantially. One approach to avoid these landscapes in the quantum setting is to use structured initial guesses, such as those adopted in quantum simulation. Another possibility is to use pre-training segment by segment, which was an early success in the classical setting[48,50]. These or other alternatives must be studied if these ansatze are to be successful beyond a few qubits.

## Methods

We explicitly show the expectation value of the gradient is 0 and that under our assumptions the variance decays exponentially in the number of qubits. By our definition of RPQCs, we have that for any specified direction $\partial_k E$, both $U_-$ and $U_+$ are independently distributed and either $U_-$ or $U_+$ match the Haar distribution up to at least the second moment (they are a 2-design). The assumption of independence is equivalent to

$$p(U) = \int dU_+ p(U_+) \int dU_- p(U_-) \\ \times \delta(U_+ U_- - U), \tag{8}$$

which allows us to rewrite the expression as

$$\langle \partial_k E \rangle = i \int dU_- p(U_-) \mathrm{Tr}\left\{ \rho_- \times \int dU_+ p(U_+) \left[ V, U_+^{\dagger} H U_+ \right] \right\}. \tag{9}$$

We will utilize explicit integration over the unitary group with respect to the Haar measure, which up to the first moment can be expressed as[51]

$$\int d\mu(U)\, U_{ij} U_{km}^{\dagger} = \int d\mu(U)\, U_{ij} U_{mk}^{*} = \frac{\delta_{im}\delta_{jk}}{N}, \tag{10}$$

where $N$ is the dimension of the space, typically $2^n$ for $n$ qubits. Using this expression, one may readily verify that

$$M = \int d\mu(U)\, U O U^{\dagger} = \frac{\mathrm{Tr}O}{N} I, \tag{11}$$

which we use in the following. Now, making use of the assumption that either $U_+$ or $U_-$ matches the Haar measure up to the first moment (it is a 1-design), we first examine the case where $U_-$ is at least a 1-design and find that

$$\langle \partial_k E \rangle = i \int d\mu(U_-) \mathrm{Tr}\left\{ \rho_- \times \left[ V, \int dU_+ p(U_+) U_+^{\dagger} H U_+ \right] \right\} \\ = \frac{i}{N} \mathrm{Tr}\left\{ \left[ V, \int dU_+ p(U_+) U_+^{\dagger} H U_+ \right] \right\}, \\ = 0 \tag{12}$$

where we have defined $\rho_- = U_- |0\rangle\langle 0| U_-^{\dagger}$ and used the fact that the trace of a commutator of trace class operators is zero. In the second case, where we assume

$U_+$ is at least a 1-design,

$$\langle \partial_k E \rangle = i \int dU_- p(U_-) \mathrm{Tr}\left\{ \rho_- \int d\mu(U_+) \left[ V, U_+^{\dagger} H U_+ \right] \right\} \\ = i \frac{\mathrm{Tr}H}{N} \int dU_- p(U_-) \mathrm{Tr}\{ \rho_- [V, I] \} \\ = 0. \tag{13}$$

An advantage of the explicit polynomial formulas are that they allow an analytic calculation of the variance as well, which allows precise specification of the coefficient in Levy's lemma. In cases where the integrals depend on up to two powers of elements of $U$ and $U^*$, one may make use of the elementwise formula[51]

$$\int d\mu(U)\, U_{i_1 j_1} U_{i_2 j_2} U_{i_1' j_1'}^{*} U_{i_2' j_2'}^{*} = \frac{\delta_{i_1 i_1'} \delta_{i_2 i_2'} \delta_{j_1 j_1'} \delta_{j_2 j_2'} + \delta_{i_1 i_2'} \delta_{i_2 i_1'} \delta_{j_1 j_2'} \delta_{j_2 j_1'}}{N^2 - 1} \\ - \frac{\delta_{i_1 i_1'} \delta_{i_2 i_2'} \delta_{j_1 j_2'} \delta_{j_2 j_1'} + \delta_{i_1 i_2'} \delta_{i_2 i_1'} \delta_{j_1 j_1'} \delta_{j_2 j_2'}}{N(N^2 - 1)}. \tag{14}$$

The variance of the gradient is defined by

$$\mathrm{Var}[\partial_k E] = \langle (\partial_k E)^2 \rangle, \tag{15}$$

as we have seen above that $\langle \partial_k E \rangle = 0$. Through use of the above formula for integration up to the second moment of the Haar distribution, one may evaluate this expression in 3 separate cases. For simplicity and relevance, we evaluate them in the asymptotic case including only the dominant contribution as determined by the inverse dimension.

In the case where $U_-$ is a 2-design but not $U_+$,

$$\mathrm{Var}[\partial_k E] \approx \frac{2\mathrm{Tr}(\rho^2)}{N^2 - 1} \mathrm{Tr}\langle H_u^2 V^2 - (H_u V)^2 \rangle_{U_+} = -\frac{\mathrm{Tr}(\rho^2)}{2^{2n} - 1} \mathrm{Tr}\langle [V, H_u]^2 \rangle_{U_+}, \tag{16}$$

where $H_u = u^{\dagger} H u$ and we have defined the notation $\langle f(u) \rangle_{U_x}$ to mean the average over $u$ sampled from $p(U_x)$. In the case where $U_+$ is a 2-design but not $U_-$,

$$\mathrm{Var}[\partial_k E] \approx \frac{2\mathrm{Tr}(H^2)}{N^2 - 1} \mathrm{Tr}\langle \rho_u^2 V^2 - (\rho_u V)^2 \rangle_{U_-} = -\frac{\mathrm{Tr}(H^2)}{2^{2n} - 1} \mathrm{Tr}\langle [V, \rho_u]^2 \rangle_{U_-}, \tag{17}$$

where $\rho_u = u \rho u^{\dagger}$. Finally in the case where both $U_+$ and $U_-$ are 2-designs

$$\mathrm{Var}[\partial_k E] \approx \frac{1}{2^{(3n-1)}} \mathrm{Tr}(H^2) \mathrm{Tr}(\rho^2) \mathrm{Tr}(V^2). \tag{18}$$

In all cases, the exponential decay of the gradient as a function of the number of qubits is evident.

## Data availability

Data used to generate the above figures are available upon request from the authors.

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

## Acknowledgements

The authors thank Craig Gidney for helpful comments on the manuscript.

## Author contributions

J.R.M., S.B., V.N.S., R.B., and H.N. contributed to the formulation of ideas, calculations, and writing of the manuscript.

## Additional information

**Competing interests:** The authors declare no competing interests.

