## [Peer Review File · Nature Communications]

Reviewers' comments:

Reviewer #1 (Remarks to the Author):

I strongly recommend that this paper not be accepted. While the authors cite a long list of references to papers which optimize a parameterized quantum circuit, very few of these papers do anything interesting. Most of the papers on quantum chemistry cited study only very small molecules, for example, and almost all completely neglect any serious scaling study. While some papers do study larger systems, these papers all employ circuits other than random quantum circuits.

So, despite the best attempts of the authors to claim otherwise, there is very scant evidence that optimizing such parametrized quantum circuits actually is useful for anything, especially in the case of random circuits. The authors now (after having themselves written many such papers themselves) claim that they have made a surprising advance: we should not expect that random circuits work well! Of course, this was not at all surprising. The basic ideas in this paper are all quite obvious and were certainly known to me as well as others beforehand. However, most people who knew such ideas in advance didn't go writing papers about random circuits because (a) it was clear it wouldn't work well and (b) if we did try simulations, we saw that it didn't scale.

In essence, the authors have created a field out of nothing and then have finally realized why it was a bad idea. The initial papers should not have been published in good journals (perhaps not anywhere) and this one certainly should not be published in a good journal.

Reviewer #2 (Remarks to the Author):

This article makes an obvious but important observation: the gradient of the cost functions used to "train" quantum circuits to perform certain tasks is exponentially small in the number of qubits. This follows from some well-known results in the field (Levy's lemma) and a simple calculation based on a natural and quite general parameterization of the circuits. The authors also make the important and perhaps less obvious point that the cost of computing the "quantum gradient" scales exponentially badly. The take away message from this is that tools like the "variational quantum eigensolver" must have a very good starting ansatz or they are likely to be useless. (One might argue that this indeed *makes* them useless...)

The results are quite clearly correct, at least in spirit and interpretation. I have not closely checked the exact calculations, as I assume that the authors carried these out correctly since they are somewhat routine. The scaling arguments that drive the main conclusions are undoubtedly correct, so there is little point in checking for factors of 2.

In spite of the simplicity and indeed obviousness of these results, I think that it is important to publish this work in Nature Comms. The reason is that it is an important observation that has seemingly gone unnoticed by the community, or at the very least they have ignored the issue. Publishing a challenge to the existing paradigm of how we propose to use NISQ devices is important, and I hope this work forces other researchers to pay closer attention to the actual claims in their papers, and to remove the hype about quantum advantages in their experiments.

Reviewer #3 (Remarks to the Author):

Referee report for Nature Coms on ``Barren Plateaus...'' by Google.

The paper "Barren plateaus in quantum neural network training landscapes" by McClean et al addresses an extremely important and timely subject: the potential for finding near-term

applications for quantum processors that outperform classical computation. In this case they analyze an approach based on hybrid quantum-classical approaches to problems in the simulation of quantum systems. The paper is short and simple but also to the point. I believe this spells bad news for the generic scalability of hybrid quantum-classical approaches and the authors could say this a bit more strongly. Overall I think this paper is a very important contribution and has the potential for high impact, as appropriate for a journal such as Nat Comms. However, it would benefit from some further clarifications as per the notes below.

The authors make the assumption that either U_- or U_+ or both are sufficiently random that they match the Haar distribution up to the second moment. I have trouble seeing why this is a reasonable assumption for any k , especially when k is close to 1 or L given that the authors give no specification for the structure of the Hermitian operators V_k . (I understand that their numerical examples suggest that it is a reasonable assumption as they seem to be considering a case with $k=1$, for $\theta_{\{1,1\}}$, if I have understood their notation correctly. Is that the case?) For example, if the V_k generate primitive elementary gates on qubits or qubit pairs, then when k is small, such small sequences of gates will not realize even 1-design across the full set of qubits. Could the authors specify what kinds of assumptions they are making on the nature of V_k to validate this assumption and how this relates to the current best practices for choosing RPQCs that are useful for the applications that motivate the study? For example, looking at figure 4, the variance is not too small for shallow depth values of L . For the applications of interest is there a good reason to choose large L values?

For Fig 3, the data here is said to represent the case where the number of layers is a "modest linear function" of the number of qubits. What did the authors choose for this modest linear function and how does it relate to relative design choices normally considered in VQE or QAOA type applications?

In Fig. 4 the authors note how the circuits saturate at a 2-design value. Could the authors include the calculated 2-design value (as a function of the number of qubits) and perhaps plot it on Fig 4 for reference?

Reviewers' comments:

Reviewer #1 (Remarks to the Author):

I strongly recommend that this paper not be accepted. While the authors cite a long list of references to papers which optimize a parameterized quantum circuit, very few of these papers do anything interesting. Most of the papers on quantum chemistry cited study only very small molecules, for example, and almost all completely neglect any serious scaling study. While some papers do study larger systems, these papers all employ circuits other than random quantum circuits.

Reviewer 1 seems to be a bit confused by the context and results of the work presented, and their comments suggest that they do not distinguish between the authors' previous works on variational algorithms and random quantum circuits, which were studied in entirely different contexts. In fact we are advocating for the papers that study larger systems using circuits other than random quantum circuits in order to prevent the trend of parametric random quantum circuits from spreading further than it has. Note that the authors' previous works on variational algorithms do not use parametric **random** circuits. We have added the following sentence to the introduction in order to emphasize this distinction: "This is in contrast to the previous proposals, such as the variational quantum eigensolver [2,3,9], which used parametrized structured circuits inspired by the problem at hand, such as unitary coupled cluster."

So, despite the best attempts of the authors to claim otherwise, there is very scant evidence that optimizing such parametrized quantum circuits actually is useful for anything, especially in the case of random circuits. The authors now (after having themselves written many such papers themselves) claim that they have made a surprising advance: we should not expect that random circuits work well!

Again, no author on this paper has ever advocated for the use of random parameterized circuits in the context of variational algorithms. It is true that some of the authors have done work on random circuits in entirely different contexts, for which they are still quite useful, and the reviewer seems to have incorrectly conflated the two application contexts. Moreover, there is indeed evidence that structured parameterized circuits can be successful in the context of chemistry with both the unitary coupled cluster or shortcutted adiabatic ansatz, see Refs [2,3,9].

Of course, this was not at all surprising. The basic ideas in this paper are all quite obvious and were certainly known to me as well as others beforehand. However, most people who knew such ideas in advance didn't go writing papers about random circuits because (a) it was clear it wouldn't work well and (b) if we did try simulations, we saw that it didn't scale.

While the reviewer believes they knew the result beforehand, the proliferation of publications which use random quantum circuits for quantum machine learning, making learning impossible, suggests that this is not as common of knowledge as they might believe.

In essence, the authors have created a field out of nothing and then have finally realized why it was a bad idea. The initial papers should not have been published in good journals (perhaps not anywhere) and this one certainly should not be published in a good journal.

We disagree with this statement. We are advocating for an important strategy aspect of a field we believe to be absolutely crucial for applications of near-term quantum devices. We in no way believe that parameterized circuits are a bad idea, nor did we suggest using random circuits in them, as our response above should make clear

Furthermore, in order to prevent similar misunderstandings, we have added the following text to the introduction of our work to provide better context and clarification about the distinction between applications of random quantum circuits and our work:

“While our results highlight the importance of avoiding random initialization in parametric circuit approaches, they do not discount the value of random quantum circuits in other applications such as information security or demonstrations of quantum supremacy.”

Reviewer #2 (Remarks to the Author):

*This article makes an obvious but important observation: the gradient of the cost functions used to "train" quantum circuits to perform certain tasks is exponentially small in the number of qubits. This follows from some well-known results in the field (Levy's lemma) and a simple calculation based on a natural and quite general parameterization of the circuits. The authors also make the important and perhaps less obvious point that the cost of computing the "quantum gradient" scales exponentially badly. The take away message from this is that tools like the "variational quantum eigensolver" must have a very good starting ansatz or they are likely to be useless. (One might argue that this indeed *makes* them useless...)*

The results are quite clearly correct, at least in spirit and interpretation. I have not closely checked the exact calculations, as I assume that the authors carried these out correctly

since they are somewhat routine. The scaling arguments that drive the main conclusions are undoubtedly correct, so there is little point in checking for factors of 2.

In spite of the simplicity and indeed obviousness of these results, I think that it is important to publish this work in Nature Comms. The reason is that it is an important observation that has seemingly gone unnoticed by the community, or at the very least they have ignored the issue. Publishing a challenge to the existing paradigm of how we propose to use NISQ devices is important, and I hope this work forces other researchers to pay closer attention to the actual claims in their papers, and to remove the hype about quantum advantages in their experiments.

We are grateful to Reviewer #2 for their careful reading of our work and appreciation of the importance of this result in the context of the current climate in the field of NISQ application development. The attention Nat Comm brings to this result will help to steer researchers in more fruitful directions and increase the probability of a near-term application win on quantum devices.

Reviewer #3 (Remarks to the Author):

Referee report for Nature Coms on ``Barren Plateaus...'' by Google.

The paper "Barren plateaus in quantum neural network training landscapes" by McClean et al addresses an extremely important and timely subject: the potential for finding near-term applications for quantum processors that outperform classical computation. In this case they analyze an approach based on hybrid quantum-classical approaches to problems in the simulation of quantum systems. The paper is short and simple but also to the point. I believe this spells bad news for the generic scalability of hybrid quantum-classical approaches and the authors could say this a bit more strongly. Overall I think this paper is a very important contribution and has the potential for high impact, as appropriate for a journal such as Nat Comms. However, it would benefit from some further clarifications as per the notes below.

We thank Reviewer #3 for their kind comments about the timeliness and importance of this work. While we believe this clarifies a challenge for hybrid quantum-classical methods, the authors remain optimistic about their prospects given the correct strategies.

The authors make the assumption that either U_- or U_+ or both are sufficiently random that they match the Haar distribution up to the second moment. I have trouble seeing why this is a reasonable assumption for any k , especially when k is close to 1 or L given that the authors give no specification for the structure of the Hermitian operators V_k . (I understand that their numerical examples suggest that it is a reasonable assumption as they seem to be considering a case with $k=1$, for $\theta_{1,1}$, if I have understood their

notation correctly. Is that the case?) For example, if the V_k generate primitive elementary gates on qubits or qubit pairs, then when k is small, such small sequences of gates will not realize even 1-design across the full set of qubits.

The Reviewer identifies an aspect that could be confusing for potential readers. The important part of the assumption is that surprisingly, only at least one of U_- or U_+ needs to be sufficiently random. Meaning the circuit which follows the point at which the derivative is taken is as important as the circuit preceding it. So for an unintuitive case, such as $k=1$, where one might imagine taking the derivative at the beginning of the circuit should always have substantial effect, in fact the randomizing effects of the circuit that follow this point are enough to ruin the gradient of almost all observables. This is one reason we chose to highlight the point $k=1$ in our numerical studies to help guide peoples intuition.

As a result, the structure of the V_k locally has very little impact, which is why our proof results don't really depend on them. The aspect in which the V_k structure matters is more global in a sense that it only determines the rate at which one can randomize the circuit. The following text has been added to the paper to clarify these important points:

“From these results, it is clear that only either U_+ or U_- needs to be sufficiently random to poison the gradient for the remainder of the circuit. For example, while it is somewhat unintuitive, even the first element of a circuit, $k=1$, will have a vanishing gradient due to the circuit following it, U_+ . Additionally, we see that there is no detailed dependence on the structure of V_k , other than the rate at which they help randomize the circuit, determining at what depth one expects to find an approximate 2-design.”

Could the authors specify what kinds of assumptions they are making on the nature of V_k to validate this assumption and how this relates the current best practices for choosing RPQCs that are useful for the applications that motivate the study? For example, looking at figure 4, the variance is not too small for shallow depth values of L . For the applications of interest is there a good reason to choose large L values?

As mentioned above, we make essentially no assumptions on V_k locally, other than it is a hermitian operator. The more global consequence that involves V_k , is given the structure of V_k , how much depth is required to randomize the circuit to a sufficient degree. In our numerical simulations we limit both the locality and connectivity of V_k in order to mimic performance on a near-term device. This increases the depth required to see this effect, but we believe more accurately represents the likely scenarios on a near-term device. This restriction we make allows for some amount of depth before the damaging effects of the randomization really kick in.

This brings up the mentioned topic of best practices for choosing RPQCs for applications of interest. Our primary recommendation is to not randomly initialize and use structured ansatz. If one insists on using a random model or has no prior guess, our circuits imply that a reasonable

course of action is to choose a circuit that is somewhere between a 1- and 2- design, before it reaches the 2-design limit. While we mention this to a cursory degree in the text, we would prefer not to advocate the use of random circuits directly in the text without further refining this idea, to avoid leading readers astray.

For Fig 3, the data here is said to represent the case where the number of layers is a “modest linear function” of the number of qubits. What did the authors choose for this modest linear function and how does it relate to relative design choices normally considered in VQE or QAOA type applications?

While we did not try to precisely quantify this in the current study, the empirical estimate is about $10 * n_{\text{qubits}}$. This is not a value we chose, but rather one that results from random circuits on a 1-d arrangement of qubits, designed to have slow convergence to a 2-design limit, or rather slower than from higher dimensional connectivities. Given that this assumed a linear connectivity of the device and 1- and 2-local V_k , this effect may be reached much faster than this in a real application. For context, a single pass of a unitary coupled cluster circuit for chemistry will typically have depth of at least $O(N^3)$ or $O(N^4)$. Hence, if one used the structure of the UCC ansatz, but randomly initialized the angles for more than a few qubits this problem would become quite relevant. We believe this comparison could be fruitful to future readers as well, so we add the following statement to tie our results more tightly to near-term applications:

“Empirically for our linear connectivity, we see that value is about $10 n$ where n is the number of qubits, following the expected scaling of $O(n^{1/d})$ where d is the dimension of the connectivity. For empirical reference, the expected gate depth in a chemistry ansatz such as unitary coupled cluster is at least $O(n^3)$, meaning that if the initial parameters were randomized, this effect could be expected on less than 10 orbitals, a truly small problem in chemical terms.”

In Fig. 4 the authors note how the circuits saturate at a 2-design value. Could the authors include the calculated 2-design value (as a function of the number of qubits) and perhaps plot it on Fig 4 for reference?

We have added the suggested line to figure 4 based on our analytic expressions for the asymptotic values of the observables and thank the Reviewer for the suggestion. We have also simplified the expressions and clarified the asymptotic order used to evaluate them. These simpler approximations are more than sufficient to determine the values given the exponential growth of dimension.

In addition to the comments made by reviewers, we have made several edits to the structure of the draft to conform with editorial requests, including a reference free introduction, separation

into a methods section, and other heading changes. These changes are highlighted in a light blue color in the revised manuscript.

In conclusion, we remain confident that dissemination of these results to the broad audience that Nature Communications enjoys will help steer the field of near-term applications on quantum computers in a better direction, which is crucial to the possibility of showing a quantum advantage for practical applications on near-term devices. Again, we thank the Reviewers and editorial staff for their work in improving our manuscript.

Sincerely on behalf of all authors,

Jarrod McClean
Senior Research Scientist
Google Quantum AI Lab

REVIEWERS' COMMENTS:

Reviewer #3 (Remarks to the Author):

My apologies for the delay.

I agree with the authors that referee 1 is missing the point. The responses to my report are satisfactory and I recommend publication without further delay.